# Beyond Language:
# Empowering Unsupervised Machine Translation with Cross-modal Alignment

## Abstract

Unsupervised machine translation (UMT) has achieved notable performance without any parallel corpora in recent years. Nevertheless, aligning the source language with the target language in the latent space remains a challenge for UMT. While different languages may exhibit variations in their textual representations, they often share a common visual description. Taking inspiration from this, in this paper, we propose a novel unsupervised multi-modal machine translation method using images as pivots to align different languages. Specifically, we introduce cross-modal contrastive learning to achieve sentence-level and token-level alignment. By leveraging monolingual image-text pairs, we align both the source and target languages in a shared semantic space using images as intermediaries, thus achieving source-to-target alignment. Experimental results demonstrate that our approach can effectively learn the source-to-target alignment with monolingual data only and achieves significant improvements over state-of-the-art methods.

## 1 Introduction

Neural machine translation (NMT) (Kalchbrenner & Blunsom, 2013; Sutskever et al., 2014) has emerged as the prevailing paradigm for machine translation with its remarkable performance. As a data-driven approach, the success of NMT often depends on the availability of extensive parallel corpus. When confronted with limited data, NMT models experience a significant drop in effectiveness. With the existence of over 7000 languages worldwide, creating a substantial amount of bilingual parallel corpora is impractical. Consequently, translation under low-resource scenarios poses a considerable challenge for NMT systems.

To address this problem, researchers have made efforts in the field of Unsupervised Machine Translation (UMT). UMT aims to translate text from a source language to a target language without any parallel corpora for training. A representative category of methods (Lample et al., 2018a;c;b; Conneau & Lample, 2019; Song et al., 2019) achieve this with three essential components: Language Modeling, Initialization, and Iterative Back-translation. Language modeling refers to train the model on large-scale monolingual corpora to learn how sentences should read in different languages. Initialization serves as a prior for the expected solution space, jump-starting the following process by providing the model with rudimentary translation ability. After initialization, back-translation is leveraged to iteratively generate pseudo-parallel corpora, allowing for the source-to-target alignment. As discussed in (Lample et al., 2018c; Huang et al., 2020), initialization, as the start of back-translation, determines the translation ability to which back-translation can ultimately iterate. Thus, the performance of UMT systems is strongly rely on proper initialization.

In recent years, an increasing number of works (Nakayama & Nishida, 2017; Li et al., 2020; Su et al., 2019; Huang et al., 2020; Fei et al., 2023) have introduced the visual modality into UMT, leading to the emergence of Unsupervised Multi-modal Machine Translation (UMMT). The visual modality, as a language-agnostic signal, has the potential to align the same semantic representations of different languages within a common space. Additionally, monolingual image-text pairs are abundant and easily accessible on social networks. Unlike parallel corpora, such data only requires annotations from monolingual speakers, eliminating the need for bilingual experts. For example, Su et al. (2019) fused visual modality with text for disambiguation. Huang et al. (2020) leveraged

an image caption model to generate pseudo-parallel sentences, facilitating data augmentation. However, it's worth noting that these studies did not fully addressed the core of UMT and primarily focus on disambiguation and data.

Therefore, we propose a novel unsupervised multi-modal method to achieve better initialization. Our method semantically aligns source-target languages into a shared latent space through contrastive learning, using images as pivots. Specifically, we introduce a sentence-level contrastive learning objective to learn coarse-grained alignment and a token-level objective to achieve fine-grained alignment. This approach ensures that if the semantics of the source and target languages are similar, their representations will be close in the shared space, which enables improved initialization, resulting in a model with good translation capabilities even before back-translation. Experiments and analysis demonstrate that our method consistently outperforms both text-only and multi-modal baselines and effectively achieves source-to-target alignment, initializing the model with good translation ability before back-translation. Furthermore, our model exhibits improvements on the out-of-domain dataset, showcasing its generalization capabilities.

## 2 BACKGROUND

### 2.1 NEURAL MACHINE TRANSLATION

NMT systems are typically based on the encoder-decoder framework. Given a parallel sentence pair $\langle \mathbf{x}, \mathbf{y} \rangle$, where $\mathbf{x} = (x_1, ..., x_n)$ represents the source sentence and $\mathbf{y} = (y_1, ..., y_m)$ represents the target sentence. The model learns translation from $\mathbf{x}$ to $\mathbf{y}$ by minimizing the cross-entropy loss:

$$\mathcal{L}_{\text{CE}} = -\sum_{i=1}^{|\mathbf{y}|} \log P(y_i | \mathbf{y}_{<i}, \mathbf{x}). \tag{1}$$

### 2.2 UNSUPERVISED MACHINE TRANSLATION

In this section, we will introduce the basic paradigm of UMT. It can be divided into three main components: Language Modeling, Initialization and Iterative Back-translation.

**Language Modeling** Language modeling aims to develop a monolingual probabilistic generation model, which entails understanding how to comprehend and generate sentences. A commonly employed training approach is the denoising autoencoder (DAE), where the model is trained to reconstruct its input from a noisy version. Building upon the DAE framework, several improvements have been proposed. For example, Lample et al. (2018a) utilizes the word dropout and random permutation, XLM Conneau & Lample (2019) leverages the cross-lingual pre-training method, and MASS Song et al. (2019) utilizes a span-based masking strategy for sequence-to-sequence learning.

In UMT, a common framework employs a parallel structure in which there are two encoders and decoders for both the source and target languages, as shown in Figure 1 (Stage 1). During training, the $S \to S$ (source-source) and $T \to T$ (target-target) directions are trained simultaneously. We denote $\mathcal{D}_x = \{\mathbf{x}_i\}_{i=1}^{M_x}$ and $\mathcal{D}_y = \{\mathbf{y}_i\}_{i=1}^{M_y}$ as two monolingual datasets of the source and target languages, respectively. Noise $\delta()$ is added to both $\mathbf{x}$ and $\mathbf{y}$ to create noisy input sentences $\delta(\mathbf{x})$ and $\delta(\mathbf{y})$. The cross-entropy loss between $\mathbf{x}$ and $\delta(\mathbf{x})$ is defined as:

$$\mathcal{L}_{\text{LM}} = -[\sum_{i=1}^{|\mathbf{x}|} \log P_{S \to S}(x_i | \mathbf{x}_{<i}, \delta(\mathbf{x})) + \sum_{i=1}^{|\mathbf{y}|} \log P_{T \to T}(y_i | \mathbf{y}_{<i}, \delta(\mathbf{y}))]. \tag{2}$$

**Initialization** The initialization equips the model with coarse-grained translation ability, jump-starting the iterative back-translation process. Specifically, initialization serves as the starting point for iterative back-translation, and its quality determines the final translation quality of the model. Klementiev et al. (2012) used a provided bilingual dictionary, Lample et al. (2018a;c) initialized the model with word-by-word translation ability using a bilingual dictionary inferred in an unsupervised way (Conneau et al., 2018b).

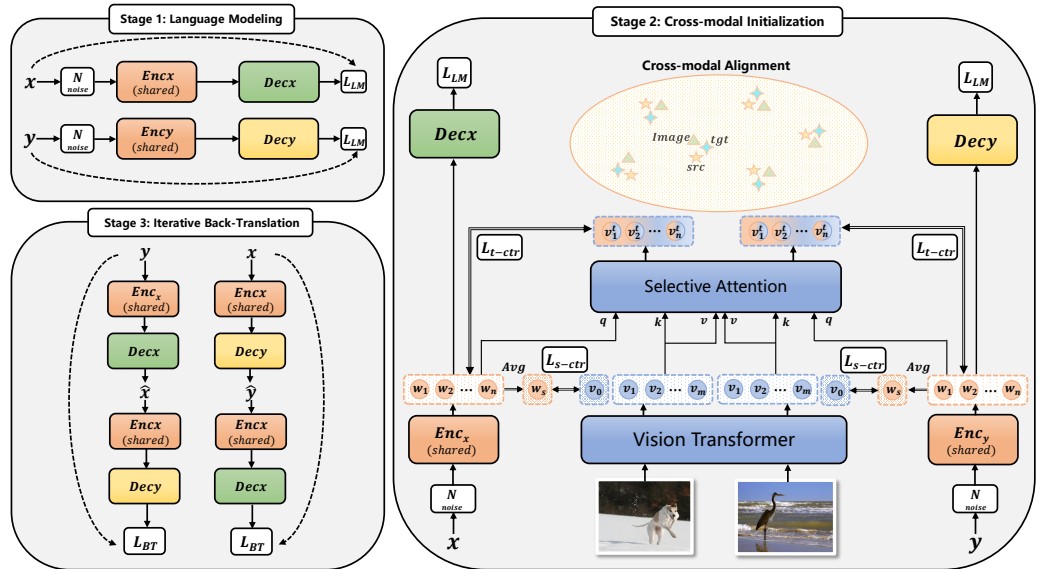

Figure 1: Overview of our proposed model. $Enc_x$, $Dec_x$, $Enc_y$, $Dec_y$ represents the source encoder, source decoder, target encoder and target decoder, respectively. The source encoder and target encoder share the parameters.

**Iterative Back-translation**    Iterative back-translation is a method proposed to automatically generate pseudo parallel sentences for source-to-target alignment. As depicted in Stage 3 of Figure 1, translation model on $S \rightarrow T$ can be obtained by recombining the obtained source encoder and target decoder. The $T \rightarrow S$ translation model can be get in the similar way. These two models continuously generating pseudo-parallel data to iteratively improve the translation performance. In detail, $\mathbf{x}$ is initially fed into the source encoder to produce $\hat{\mathbf{y}}$ via the output of target decoder. Similarly, $\mathbf{y}$ is input into the target encoder to obtain $\hat{\mathbf{x}}$ through the source decoder. The pseudo-translation results, $\hat{\mathbf{x}}$ and $\hat{\mathbf{y}}$, are generated using beam search. In this way, with pseudo parallel corpus $(\mathbf{x}, \hat{\mathbf{y}})$ and $(\hat{\mathbf{x}}, \mathbf{y})$, the model is trained by minimizing the cross-entropy loss between pseudo parallel sentences:

$$\mathcal{L}_{\mathrm{BT}} = -[\sum_{i=1}^{|\mathbf{x}|} \log P_{T \rightarrow S}(x_i | \mathbf{x}_{<i}, \hat{\mathbf{y}}) + \sum_{i=1}^{|\mathbf{y}|} \log P_{S \rightarrow T}(y_i | \mathbf{y}_{<i}, \hat{\mathbf{x}})]. \tag{3}$$

## 3    INITIALIZATION WITH CROSS-MODAL ALIGNMENT

As stated in Section 2.2, the model learns source-to-target mapping through back-translation. Therefore, as the starting of back-translation, initialization should align its objectives as closely as possible with back-translation. To address this, in this section, we present our proposed cross-modal alignment method to establish the initial source-to-target mapping. The method consists of two parts, coarse-grained sentence-level contrastive learning and fine-grained token-level contrastive learning (Yang et al., 2022), which we will describe in detail below.

### 3.1    MODEL FRAMEWORK

Our model is built upon the introduced framework in Section 2.2, which consists of two encoders and two decoders. For the source language $x$ and target language $y$, we have monolingual image-text pairs, which contains $\{(\mathbf{x}_i, \mathbf{i}_i)\}_{i=1}^{M}$ and $\{(\mathbf{y}_i, \mathbf{i}_i)\}_{i=1}^{N}$, respectively. Notably, images of different languages do not duplicate. For cross-modal alignment, we propose a cross-modal contrastive module, which contains sentence-level and token-level objectives. We will illustrate the alignment of the source language as an example, and the target language follows the same procedure.

The *source encoder* and *target encoder* consist of $N$ Transformer encoder layers. In order to train a shared latent semantic space, the two encoders share parameters with each other. For the input sentence $\mathbf{x} = (x_1, ..., x_n)$, the output of *encoder* is denoted as $\mathbf{w} = (w_1, ..., w_n)$. The *decoders* consists of $N$ Transformer decoder layers.

For the *image encoder*, we use Vision Transformer (ViT) (Dosovitskiy et al., 2021) to extract visual features. ViT encodes the image to a sequence $\mathbf{v} = (v_0, v_1, ..., v_m)$, where $v_0$ is the special `[class]` token and the others are the representation of image patches.

## 3.2 SENTENCE-LEVEL CONTRASTIVE LEARNING

The key idea of contrastive learning (Sohn, 2016) is to bring representations of corresponding pairs closer together while pushing irrelevant pairs farther apart. We first perform coarse-grained alignment at the sentence-level. We average the encoder output of text as the text sentence-level representation and take the special `[class]` token $v_0$ of ViT as the global feature of the image.

$$\mathbf{w}^s = \frac{1}{n} \sum_{i=1}^{n} w_i, \; \mathbf{v}^s = v_0. \tag{4}$$

In this way, we can encode a image-text batch of size $B$ to $\mathbf{W}^s = \{\mathbf{w}_i^s\}_{i=1}^{B}$ and $\mathbf{V}^s = \{\mathbf{v}_i^s\}_{i=1}^{B}$ respectively. In this batch, $(\mathbf{w}_i^s, \mathbf{v}_i^s)$ are positive examples while $(\mathbf{w}_i^s, \mathbf{v}_j^s)(i \neq j)$ are negative ones. In order to push the positive examples closer while keeping the negative examples away from each other, we can use infoNCE loss (van den Oord et al., 2019) to achieve this goal:

$$\mathcal{L}_{\text{s-ctr}}(\mathbf{x}, \mathbf{i}) = -\sum_{i=1}^{M} [\log \frac{\exp(s(\mathbf{w}_i^s, \mathbf{v}_i^s)/\tau)}{\sum_{j=1}^{M} \exp(s(\mathbf{w}_i^s, \mathbf{v}_j^s)/\tau)} + \log \frac{\exp(s(\mathbf{v}_i^s, \mathbf{w}_i^s)/\tau)}{\sum_{j=1}^{M} \exp(s(\mathbf{v}_i^s, \mathbf{w}_j^s)/\tau)}] \tag{5}$$

where $s()$ is the cosine similarity $s(a, b) = a^\top b / \|a\| \|b\|$ and $\tau$ is the temperature hyper-parameter.

## 3.3 TOKEN-LEVEL CONTRASTIVE LEARNING

Through sentence-level contrastive learning, we have learned coarse-grained alignment between text and images modalities, and furthermore, we learn fine-grained alignment through token-level contrastive learning to improve the performance of the model.

In token-level contrastive learning, we focus on each sentence and its corresponding image. We encode them into two sequences $\mathbf{w} = (w_1, ..., w_n)$ and $\mathbf{v} = (v_1, ..., v_m)$. Since there is sequence length inconsistency between the text and the image sequences and there is always redundant information in the global feature of images, we use selective attention (Li et al., 2022) to standardize sequence lengths and filter out irrelevant information. We denote $\mathbf{w}, \mathbf{v}, \mathbf{v}$ as the query, key and value of selective attention, respectively.

$$\mathbf{v}^t = \text{Softmax} \left( \frac{(W_Q \cdot \mathbf{w})(W_K \cdot \mathbf{v})^\top}{\sqrt{d_k}} \right) (W_V \cdot \mathbf{v}), \tag{6}$$

where $W_Q$, $W_K$ and $W_V$ are learnable matrix parameters. Therefore, we can get $\mathbf{w} = (w_1, ..., w_n)$ and $\mathbf{v}^t = (v_1, ..., v_n)$. The positive examples are $(w_i, v_i^t)$ and negative examples are $(w_i, v_j^t)(i \neq j)$, the loss function of token-level contrastive learning can be defined as follows:

$$\mathcal{L}_{\text{t-ctr}}(\mathbf{x}, \mathbf{i}) = -\sum_{k=1}^{M} \sum_{i=1}^{|\mathbf{w}|} [\log \frac{\exp(s(w_i, v_i^t)/\tau)}{\sum_{j=1}^{|\mathbf{w}|} \exp(s(w_i, v_j^t)/\tau)} + \log \frac{\exp(s(v_i^t, w_i)/\tau)}{\sum_{j=1}^{|\mathbf{w}|} \exp(s(v_i^t, w_j)/\tau)}] \tag{7}$$

# 4 EXPERIMENTS

## 4.1 DATASETS

The data we used comes from three datasets, namely WMT News Crawl, MsCOCO (Lin et al., 2014) and Multi30K (Elliott et al., 2016). WMT News Crawl is a large-scale monolingual dataset

that includes multiple languages. We shuffle the WMT News Crawl from 2007 to 2017 and take the first 10M sentences for training. MsCOCO (Lin et al., 2014) is an English annotated image dataset. Specifically, we work with the Caption 2015 set, consisting of 121,000 image-text pairs. Following Huang et al. (2020), we translate half of the dataset into German and French. Multi30K (Elliott et al., 2016) is a benchmark dataset of multi-modal machine translation. The training and validation sets consist of 29,000 and 1,014 sentences in German, French and English with paired images. For evaluation, we assessed our model on the Test2016, Test2017, and MsCOCO test sets, which respectively contains 1,000, 1,000, and 461 instances.

## 4.2 TRAINING DETAILS

**Language Modeling**   We follow Su et al. (2019); Huang et al. (2020) to combine a 10M subset of the WMT monolingual corpus with ten times the amount of the 14.5K (half of Multi30K), resulting in a combined monolingual dataset of 10.145 million sentences. We leverage the MASS (Song et al., 2019) objective for language modeling. We mask off a contiguous span of the original sentence and ask the decoder to reconstruct the masked span. More details can be found in the original paper.

**Initialization with Cross-modal Alignment**   During the initialization stage, we utilize a dataset consisting of 75,000 monolingual image-text pairs for each language, combining half of the COCO and Multi30K datasets. Note that we ensure that the images in different languages do not overlap in this case. During this process, as the sentences output by the MASS method are segments, we additionally introduce a token mask loss to make the output sentences more fluent. We randomly mask some tokens of the input and ask the decoder to output the complete sentence.

**Iterative Back-translation**   Lastly, we train iterative back-translation on the 14.5K half of Multi30K monolingual dataset for a fair comparison with baseline systems. Notably, to enhance the model's applicability, unlike most UMMT systems (Su et al., 2019; Huang et al., 2020), during training, we do not introduce any visual modality, resulting a inference-time image-free model.

## 4.3 SYSTEM SETTINGS

Our model is based on the Transformer (Vaswani et al., 2017) architecture. Both the encoder and decoder have $N = 6$ layers. The number of attention heads is set to 4, the input embedding dimension is 512 and the feed forward embedding dimension is 1024. We apply a dropout rate of 0.3, and a label smoothing of 0.1. For optimizing, we use Adam optimizer (Kingma & Ba, 2015) and 2000 warm-up updates. The learning rate is 5e-4. Each batch contains a maximum of 4,096 tokens. During language modeling, we train our model for a total of 15 epochs.

We use ViT (Dosovitskiy et al., 2021) as the image encoder, which converts images into a 512-dimensional embedding. The output sequence length is 50, consisting of a special `[class]` token and 49 feature tokens. In cross-modal initialization and back-translation, we keep the training parameters the same as in the language modeling and implement an early stop strategy, where training is stopped if the validation loss does not decrease within 10 epochs.

For evaluation, we average the last 5 checkpoints and use beam search with a beam size of 5. We evaluate the model using multi-BLEU (Papineni et al., 2002) score computed by multi-bleu.pl[1], and the METEOR (Banerjee & Lavie, 2005) score calculated using the METEOR tool[2]. We implement our system on *fairseq*[3] (Ott et al., 2019). Our experiments are conducted on 4 NVIDIA 3090 GPUs.

## 4.4 BASELINE SYSTEMS

We compare our method with both the unsupervised text-only and the multi-modal baseline models. The text-only baselines includes: MUSE (Conneau et al., 2018a), UNMT (Lample et al., 2018a), XLM (Conneau & Lample, 2019) and MASS (Song et al., 2019). The multi-modal baselines includes: UMMT (Su et al., 2019), PVP (Huang et al., 2020), SG (Fei et al., 2023).

---

[1]https://github.com/moses-smt/mosesdecoder/blob/master-/scripts /generic/multi-bleu.perl

[2]https://github.com/cmu-mtlab/meteor

[3]https://github.com/pytorch/fairseq

Table 1: Results of UNMT systems on Multi30K Flickr2016. UMMT* and PVP* are results reimplement by Fei et al. (2023) using visual hallucination method Fang & Feng (2022)

| Models | EN→DE | | DE→EN | | EN→FR | | FR→EN | | Avg | |
|---|---|---|---|---|---|---|---|---|---|---|
| | BLEU | METEOR | BLEU | METEOR | BLEU | METEOR | BLEU | METEOR | BLEU | METEOR |
| ● *Text-only systems* | | | | | | | | | | |
| MUSE (Conneau et al., 2018a) | 15.7 | - | 5.4 | - | 8.5 | - | 16.8 | - | 11.6 | - |
| UNMT (Lample et al., 2018a) | 22.7 | - | 26.3 | - | 32.8 | - | 32.1 | - | 28.5 | - |
| XLM (Conneau & Lample, 2019) | 28.7 | 48.7 | 30.7 | 31.0 | 46.3 | 64.3 | 42.0 | 38.1 | 36.9 | 45.5 |
| MASS (Song et al., 2019) | 27.3 | 48.1 | 32.3 | 33.0 | 47.6 | 64.5 | 43.3 | 38.3 | 37.6 | 46.0 |
| ● *Multi-modal systems without image input given* | | | | | | | | | | |
| UMMT (Su et al., 2019) | 8.4 | 11.3 | 7.5 | 10.8 | 15.8 | 12.7 | 10.2 | 13.6 | 10.5 | 12.1 |
| UMMT* (Fei et al., 2023) | 15.7 | 17.7 | 19.3 | 22.7 | 30.4 | 28.4 | 31.8 | 30.4 | 24.3 | 24.8 |
| PVP (Huang et al., 2020) | 11.1 | 13.8 | 14.0 | 17.2 | 26.1 | 23.8 | 25.7 | 23.4 | 19.2 | 19.6 |
| PVP* (Fei et al., 2023) | 25.4 | 40.1 | 27.6 | 26.0 | 46.7 | 58.9 | 39.0 | 31.9 | 34.6 | 39.0 |
| SG (Fei et al., 2023) | 32.0 | 52.3 | 33.6 | 32.8 | **50.6** | 64.7 | 45.5 | 37.3 | 40.4 | 46.7 |
| Ours | **36.0** | **55.2** | **38.2** | **36.5** | 50.0 | **65.3** | **46.6** | **39.7** | **42.7** | **49.2** |

Table 2: Results on Multi30K Flickr2017 set and COCO2017 set.

| Sets | Models | EN→DE | | DE→EN | | EN→FR | | FR→EN | | Avg | |
|---|---|---|---|---|---|---|---|---|---|---|---|
| | | BLEU | METEOR | BLEU | METEOR | BLEU | METEOR | BLEU | METEOR | BLEU | METEOR |
| F17 | MASS | 22.8 | 30.3 | 27.8 | 43.5 | 42.5 | 58.8 | 38.0 | 34.8 | 32.8 | 41.8 |
| | Ours | **28.8** | **34.1** | **31.4** | **49.0** | **44.4** | **60.5** | **41.4** | **37.1** | **36.5** | **45.2** |
| C17 | MASS | 24.4 | 43.5 | 26.1 | 30.3 | 37.5 | 56.2 | 36.4 | 35.0 | 31.1 | 41.2 |
| | Ours | **27.5** | **46.7** | **27.7** | **32.8** | **39.3** | **57.9** | **40.8** | **37.2** | **33.8** | **43.6** |

## 4.5 RESULTS

We compared our model with other state-of-the-art UMT and UMMT systems. As shown in Table 1, our model demonstrates significant improvements in BLEU scores compared to the text-only models. Compared to the state-of-the-art text-only baseline MASS (Song et al., 2019), our method achieves an average BLEU score improvement of 5.1 and an average METEOR score improvement of 3.2 across the four language directions. This indicates the crucial role of cross-modal alignment.

In the second part of Table 1, all the entries represent UMMT systems, and to ensure fair comparison, they are tested without image input given. The UMMT and PVP are test without image input given. The UMMT* and PVP* are results reimplemented by Fei et al. (2023) using visual hallucination method (Fang & Feng, 2022) since they have image input in their original method. It is evident that our proposed method exhibits significant improvements in both BLEU and METEOR metrics compared to other UMMT systems. Notably, our method achieves a remarkable increase of 2.3 average BLEU and 2.5 average METEOR across the four language directions when compared to the recent state-of-the-art system SG (Fei et al., 2023). This establishes our method as the new state-of-the-art in the field of UMMT.

Additionally, We evaluated our model on the Flickr2017 and MsCOCO2017 sets of Multi30K in Table 2, which was not done by other UMMT systems. As most UMMT methods do not have open-source code, we compared our results with MASS (Song et al., 2019) on these two test sets. We achieved significant improvements as well, further confirming the effectiveness of our approach.

## 5 ANALYSIS

### 5.1 ABLATION STUDIES

We conduct ablation studies to quantify the contribution of each objective, as shown in Table 3. (1) The cross-modal initialization plays an crucial role in the model. Compare line 5 with 7, we observe a noticeable 5.1 decrease in BLEU scores across all language directions. (2) The language modeling is another important component, this step enables the model to learn better monolingual representation, which can enhance its performance in subsequent training stages. Compare line 2 with 4, the model trained with language modeling achieves 5.3 BLEU score improvements. (3)

Table 3: Ablation studies. BLEU score of different learning strategies. L: Language modeling, S: Sentence-level contrastive loss, T: Token-level contrastive loss, B: Back translation.

| ID | Model | EN→DE | DE→EN | EN→FR | FR→EN | Avg |
|----|-------|-------|-------|-------|-------|-----|
| 1 | S | 22.6 | 25.7 | 20.3 | 24.5 | 23.3 |
| 2 | S+T | 25.1 | 27.3 | 20.8 | 25.6 | 24.7 |
| 3 | L+S | 26.1 | 29.4 | 31.3 | 30.3 | 29.3 |
| 4 | L+S+T | 27.5 | 30.0 | 31.6 | 30.8 | 30.0 |
| 5 | L+B | 27.3 | 32.3 | 47.6 | 43.3 | 37.6 |
| 6 | L+S+B | 34.6 | 36.7 | 49.4 | 46.1 | 41.7 |
| 7 | L+S+T+B(Full) | **36.0** | **38.2** | **50.0** | **46.6** | **42.7** |

Table 4: Text-to-image retrieval results on Flickr2016 EN→DE.

| Models | R@1↑ | R@5↑ | R@10↑ |
|--------|------|------|-------|
| MASS | 0.3 | 1.4 | 1.7 |
| Ours | **46.6** | **75.5** | **84.2** |

Table 5: BLEU scores without back-translation. MUSE (Conneau et al., 2018b) is the initialization objective adopted by UNMT (Lample et al., 2018a).

| Model | EN→DE | DE→EN | EN→FR | FR→EN |
|-------|-------|-------|-------|-------|
| MUSE | 15.7 | 5.4 | 8.5 | 16.8 |
| MASS | 16.7 | 12.4 | 16.6 | 19.7 |
| Ours | **27.5** | **29.9** | **31.6** | **30.8** |

Futhermore, when we compare line 1, 2 with 3, 4, we can find that back-translation significantly improves translation performance (around 6 BLEU points). This highlights the important role of pseudo-parallel corpora in training. (4) Additionally, compare line 3, 6 with 4, 7, the token-level contrastive method yields an approximately 1-point gain in BLEU compared to the sentence-level one, demonstrating that fine-grained alignment leads to better results.

## 5.2 SEMANTIC ALIGNMENT

Can our model achieve semantic alignment between different languages in the latent space? To examine it, we conduct some analysis of text and image representations.

**Text-to-image Retrieval**  To validate the alignment between text and images, we compute the cosine similarity between each sentence and all the images, selecting the top-$K$ images with the highest similarity scores. The Recall@$K$ score for $K = 1, 5, 10$ is shown in Table 4. The model trained with cross-modal alignment significantly outperforms the model trained only with language modeling in terms of retrieval accuracy, proving that the contrastive learning objective enables cross-modal alignment.

**Visualization**  To gain a more intuitive understanding of the source and target representations in latent space, we utilize PCA (Principal Component Analysis) to reduce the dimensionality of the sentence-level representations from 512 to 2, and visualize them. As shown in Figure 2, our approach successfully reduces the distance between sentence representations that have similar semantics compared to the baseline model.

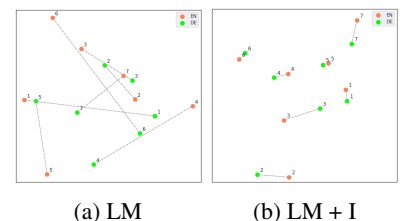

(a) LM             (b) LM + I

Figure 2: Visualization of sentence-level representations for DE and EN. (a) Language Modeling. (b) Language Modeling + Initialization. Sentences are from Multi30K Test2016 sets.

**Translation Quality**  To further analyze the effectiveness of cross-modal initialization, we compare the translation quality of the model before back-translation. MUSE (Conneau et al., 2018b) is a word-to-word translation model initialized by a inferred bilingual dictionary. As shown in Table 5, our model exhibit significant improvements compared to other baselines, even outperforming UNMT (Lample et al., 2018a) that underwent back-translation training. This illustrates

that through cross-modal contrastive learning, the model successfully acquires a common semantic space, transfers strong translation abilities from monolingual tasks, as a result, achieves a higher-quality initialization.

## 5.3 OUT-OF-DOMAIN PERFORMANCE

To further validate the generality of our method, we conduct extra experiments on the commonly used IWSLT dataset for text-only machine translation. IWSLT is a spoken language dataset that includes a variety of topics from TED talks, making it more aligned with real-world translation tasks compared to Multi30K. To accurately evaluate the out-of-domain performance of the model, unlike previous works (Fei et al., 2023), we do not introduce any additional images or employ text-to-image retrieval to find matching images. Instead, we solely rely on the existing 70K text-image pairs for cross-modal initialization and only train iterative back-translation on IWSLT.

Table 6: BLEU scores on IWSLT14 EN-DE and IWSLT17 EN-FR test sets.

| Model | EN→DE | DE→EN | EN→FR | FR→EN |
|-------|-------|-------|-------|-------|
| MASS | 22.6 | 21.9 | 33.1 | 31.9 |
| Ours | **23.3** | **22.4** | **33.2** | **32.4** |

We conducted experiments on the IWSLT14 EN-DE and IWSLT17 EN-FR datasets. The EN-DE direction includes 174K training data and 6.7K test data, while the EN-FR direction includes 236K training data and 8.5K test data. As shown in Table 6, compared to the strong text-only baseline MASS (Song et al., 2019), our method shows improvements in all four language directions, demonstrating the effectiveness of our approach on out-of-domain datasets.

## 5.4 PERFORMANCE ACROSS LINGUISTICALLY DIVERSE LANGUAGES

For English and French, there is a substantial amount of shared vocabulary, indicating a higher degree of similarity. In the case of English and German, the differences between them are relatively greater, but they still belong to the same language family. Therefore, in order to explore the effectiveness of the alignment method when applied to languages with low similarity, we chose to conduct experiments with Czech, a language that does not belong to the same language family as English. English belongs to the Indo-European language family, while Czech belongs to the West Slavic language group. As shown in Table 7, Our approach demonstrates superior performance in Czech compared to MASS (Song et al., 2019).

Table 7: Results on Multi30K EN-CS Flickr2017 set and Flickr2018 set.

| Sets | Models | EN→CS | | CS→EN | | Avg | |
|------|--------|-------|--------|-------|--------|-------|--------|
| | | BLEU | METEOR | BLEU | METEOR | BLEU | METEOR |
| F17 | MASS | 20.1 | 23.9 | 27.1 | 29.3 | 23.6 | 26.6 |
| | Ours | **24.2** | **26.4** | **30.8** | **32.2** | **27.5** | **29.3** |
| F18 | MASS | 16.1 | 21.2 | 22.3 | 27.1 | 19.2 | 24.1 |
| | Ours | **20.0** | **24.1** | **26.6** | **30.4** | **23.3** | **27.3** |

## 5.5 CASE STUDY

In this section, we make a qualitative analysis with several examples. Table 8 compares the qualitative results of the text-only MASS (Song et al., 2019) model, our model without back-translation, and the complete model. Comparing cases in two language directions, our model exhibits superior translation quality compared to MASS. For example, the term "at night" in Case 1, and "googles", "at a bus stop" in Case 2.

Additionally, it can be observed from the examples that back translation plays a crucial role in translation quality, especially in grammar. Model trained without back-translation often produces sentences with grammatical errors, such as "walks the street" in Case 1.

## 6 RELATED WORKS

**Unsupervised MT**   Unsupervised Machine Translation refers to achieving translation tasks using only monolingual corpora. Early methods (Firat et al., 2016; Chen et al., 2017; Cheng et al., 2017;

Table 8: Qualitative examples on Multi30K test sets. The red text indicates the translation error, the green text indicates the correct translations, and the *(words in brackets)* indicates the missing words.

| | Models | |
|---|---|---|
| | **Case 1 DE→EN** | |
| Ref. | SRC | Eine frau geht die straße entlang. |
| | TGT | A woman walking down the street. |
| Ours (w/o BT) | | A woman walks the street. |
| MASS | | A woman walks down the street at night. |
| Ours (Full) | | A woman walking down the street. |
| | **Case 2 FR→EN** | |
| Ref. | SRC | Un homme en costume tenant une boisson dans un gobelet marchant sur le trottoir, à côté d' un bus. |
| | TGT | A male in a suit holding a beverage in a cup walking down the sidewalk, next to a city bus. |
| Ours (w/o BT) | | A man in *(a)* costume holding a drink *(in a cup)* in a crosswalk walking on the sidewalk, near a bus. |
| MASS | | A man in a suit holding a drink in a goggles walking on the sidewalk, at a bus stop. |
| Ours (Full) | | A man in a suit holding a drink in a mug walking down the sidewalk, next to a bus. |

Johnson et al., 2017) use a third language as a pivot to achieve zero-shot translation, but such methods did not fully overcome the limitation of requiring parallel corpora. Lample et al. (2018a;c;b) propose a novel unsupervised method, which initializes the model with large-scale monolingual data and trains the source-target alignment by constructing pseudo-parallel corpora through back-translation. Subsequent works (Conneau & Lample, 2019; Song et al., 2019) follow this line by improving pre-training methods. However, as mentioned in Lample et al. (2018c), the source-target alignment is uncertain. Therefore, in this paper, we leverage visual modality and contrastive objective to learn better alignment.

**Unsupervised MMT**  Unsupervised Multi-modal Machine Translation aims to introduce visual modality to enhance UMT. Previous works (Chen et al., 2018; Su et al., 2019) fuse visual and textual information to enhance the UMT model. Another line of research is to achieve zero-shot translation with image as a pivot Nakayama & Nishida (2017); Li et al. (2020); Huang et al. (2020). However, such methods still require images as input during inference. We extend this research line and achieve better performance while eliminating the need for image inputs during inference.

**Cross-modal Contrastive Learning**  Contrastive learning van den Oord et al. (2019), as a newly self-supervised learning method, has achieved excellent performance in many tasks (Huang et al., 2021; Xu et al., 2021; Yan et al., 2021; Fei et al., 2022; Huang et al., 2022). The CLIP Radford et al. (2021) is indeed one of the notable applications of contrastive learning. It leverages cross-modal contrastive learning to align images and text, enabling zero-shot prediction. Recent studies (Ye et al., 2022; Yang et al., 2022) have indicated that cross-modal contrastive learning has achieved promising results in the field of NMT as well. Inspired by these efforts, we propose a cross-modal contrastive learning method to empower UMT systems.

# 7 CONCLUSION

In this paper, we propose a novel UMMT method that incorporates a cross-modal contrastive objective, which enables model to learn source-to-target alignment for improved initialization. Experimental results show that our method gains significant improvements over both text-only and multi-modal baseline and set a new state-of-art in UMMT. Further analysis indicates that our method successfully achieves semantic alignment of diverse language representations in the latent space. In the future, we will explore the application of our method in more low-resource scenarios.

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

## A  EFFECT OF VARIOUS NOISES

In language modeling, MASS (Song et al., 2019) has already been proven to be the most effective method. However, continuing with the MASS (Song et al., 2019) approach during initialization can result in model outputs being fragments, which contradicts the translation objective. Therefore, we introduce a token mask noise, which randomly masks the input token, and to train the model to reconstruct complete sentence outputs. To explore whether other noises with complete sentence output can achieve better results, we conduct comparative experiments on three types of noise: token deletion, token permutation, and token mask.

Table 9: BLEU scores of initialization trained with various noise.

| Noise | EN→DE | DE→EN | EN→FR | FR→EN |
|---|---|---|---|---|
| Deletion | 20.4 | 24.6 | 22.5 | 25.6 |
| Permutation | 18.6 | 21.5 | 20.0 | 20.7 |
| Mask | **27.5** | **30.0** | **31.6** | **30.8** |

As shown in Table 9, the token mask loss achieve the best result. In detail, we observe that the token mask task is easier to learn compared to the other two tasks, which is why the model can obtain better translation ability.

## B  SINGULAR VALUE GAP AND EFFECTIVE CONDITION NUMBER

In order to show how the shared representation become after our proposed method is applied, we report singular value gap and effective condition number (Dubossarsky et al., 2020) to quantitatively demonstrate the effectiveness of our method. The singular value gap provides an empirical quantification of the disparity in the complete spectral information between two embedding spaces. Meanwhile, the effective condition number gauges the degree of variation in the function's output value in response to a slight change in the input. Details about these two metrics can be found in the original paper (Dubossarsky et al., 2020).

Table 10: Singular value gaps of EN-DE and EN-FR.

| Models | EN-DE | EN-FR |
|---|---|---|
| Stage 1 | 63.3 | 4.7 |
| Stage 1+2 | 2.7 | 0.1 |

Table 11: Effective Condition Number of EN, DE and FR

| Model | EN | DE | FR |
|---|---|---|---|
| Stage 1 | 24.3 | 19.4 | 24.3 |
| Stage 1+2 | 16.7 | 17.4 | 17.1 |

As shown in Table 10 and Table 11, it can be observed that through contrastive learning, the singular value gaps in both English-German and English-French pairs significantly reduced, and the effective condition number also decreased relatively. This further emphasizes the effectiveness of our method.

## C  EXPERIMENTS ON LOW-RESOURCE SETTINGS

To further supplement our findings, we conducted experiments on English-to-German and English-to-French translation under low-resource settings, specifically using only 75,000 monolingual data samples. The experimental results in Table 12 indicate that in simulated low-resource scenarios, our method continues to yield significant improvements. In contrast, text-only method without large-scale monolingual pre-training experiences a severe performance drop, and the model convergence during training is notably slower.

Table 12: Results on 75K EN-DE and EN-FR data only.

| Sets | Models | EN→DE | | DE→EN | | EN→FR | | FR→EN | | Avg | |
|---|---|---|---|---|---|---|---|---|---|---|---|
| | | BLEU | METEOR | BLEU | METEOR | BLEU | METEOR | BLEU | METEOR | BLEU | METEOR |
| F17 | MASS | 18.3 | 35.8 | 19.7 | 23.4 | 24.0 | 41.5 | 23.9 | 27.0 | 21.5 | 31.9 |
| | Ours | **27.5** | **47.6** | **32.3** | **32.2** | **36.7** | **54.6** | **36.9** | **34.3** | **33.6** | **42.2** |

