# OpenReview forum: "Beyond Language: Empowering Unsupervised Machine Translation with Cross-modal Alignment"
_ICLR.cc/2024/Conference — Submitted to ICLR 2024_

### Official Review · Reviewer_fBBj · 2023-10-22

**Soundness:** 3 good
**Presentation:** 2 fair
**Contribution:** 3 good
**Rating:** 5
**Confidence:** 4

**Summary:**

This paper describes a method for unsupervised multi-modal machine translation (UMMT). The novelty of the method is introducing a sentence-level and token-level contrastive learning signal into UMMT.

UMMT is described in 3 stages: (1) language modeling, (2) initialization, and (3) iterative back translation. The authors describe the method as a contribution to stage (2) initialization by using noise contrastive estimation on batches of image-text/caption pairs to improve the semantic alignment of the UMMT model without bi-text between the language pairs. Experimental results for Multi30K Flickr2016, Flickr2017 and COCO2017 demonstrate that the method yields better performance across nearly all MT directions (EN<-->FR or DE).
Ablations identify that all components are useful in the final UMMT model and some analysis also discusses how the semantic space is more aligned with this method than others.

**Strengths:**

- This paper describes a novel integration of NCE-based learning to use extra image-text corpora into translation.
This will be valuable for both MT and multimodal ML researchers in generating new ideas and advancing multimodal machine translation.

- The interaction between multilingual representation alignment and multimodal representation alignment is not fully understood at present. this paper contributes to advancing this understanding.

**Weaknesses:**

- The experimental setup is arguably slightly outdated in that the paper trains a very small transformer from scratch for the task. While it is fair to do this for fair comparison to other work, the contributions of the paper could be extended and more applicable to a contemporary training setup if they used a pre-trained initialization (in lieu of stage (1) of the UMMT pipeline). This hurts the impact of the paper in its current form as it is less clear how this work could be interpreted/extended by a current reader. At a minimum, the paper should address why a pretrained model would not be appropriate here if this is the case.

- The work has minimal diligence for the handling of data for the task. 3 datasets are combined, chopped, shuffled and split into different groups and splits for each stage but this is not justified or introspected upon. It is not clear why the work takes the current course of action. Furthermore, it is not clear if different splits (i.e. more data in (1) or (2) would be more beneficial given the purported low-resource benefit of the method).

- I am concerned some of the analyses and ablations are straw men. For the retrieval task, I can see little reason why MASS should be expected to produce text-to-image alignment as this was not the intention of this model. 5.4 appears to only re-enforce prior work and I am not sure why this is a core contribution. The analysis in 5.5 appears to make conclusions from a sample of 2 outputs which is not sufficient for a qualitative or quantitative conclusion.

- ~Comparisons to other systems are not current. For text-only comparisons there is no reference to OPUS, NLLB, M2M100, MBART50-(one,many)-to-(many,one) or any recent comparator for text-only MT. This makes your results harder to contextualize. These models can be run with the compute available to the authors.~

**Questions:**

- The framing of the Initialization stage is confusing and needs more contextualization — is this optional after pre-training. what prior work uses or skips this? how was better pre-training changed the need for initialization? Furthermore, you describe Initialization somewhat circularly, in that you say that it is important because it happens and do not clearly state what the purpose of this stage is and why it is needed.

- Many uses of imprecise or hyperbolic language which is arguably inappropriate. "a certain level of translation ability" is redundant. "remarkable increase" is not justified.

- You appear to have misused "significant" as an adjective without statistically quantifying if your results are statistically significant. Please either revise or confirm the statistical significance of your results.

- Many spelling and grammar errors i.e. "langauge", "acheive", "We" in the middle of a sentence.

- Why must decoders be language specific?

- Did you consider using a [CLS] type token for the encoding to avoid needing to pool the encodings?

- Did you consider that your improvements, compared to models trained not on image-text data, may be because you are better fitting to the distribution of sequence lengths of this kind of data which other MT models are unaware of?

---

> ### Author Response · Authors · 2023-11-15
> **Response to Reviewer fBBj (1/2)**
>
> Thank you for your suggestion, but there might be some misunderstandings between us. I hope the following responses would help address your concerns.
>
> **1. Concerns on the unsupervised machine translation and multilingual machine translation**
>
> Firstly, we believe there might be some misunderstanding regarding our task. The focus of our investigation is unsupervised machine translation, where no parallel corpora are used to achieve translation from source to target. In contrast, models like NLLB and M2M100, which mentioned in the review, often rely on more extensive parallel corpora in their training data and fall within the realm of multilingual translation. Hence, the differences in training data and approaches result in distinct tasks between unsupervised machine translation and multilingual translation.
>
> Within multilingual translation tasks, there are indeed zero-shot subtasks, such as evaluating the translation performance for language pairs without parallel corpora. However, fair comparisons in such cases should involve other multilingual translation models but not unsupervised translation models.
>
> In a broader context, unsupervised machine translation explores unsupervised training methods that can be applicable to various tasks, including multilingual translation. Our approach can be implemented in many multilingual translation models, such as applying contrastive learning at the representation level to mitigate off-target issues. Nevertheless, directly comparing our method to the majority of multilingual translation models may not be entirely equitable.
>
> Additionally, due to hardware limitations (we only have 20GB 3090s), we couldn't conduct experiments on a larger scale.
> Under these computational resource constraints, even running a 3b model is challenging. We will explore further with more extensive computational resources in the future.
>
> **2. Concerns on the use of pre-trained models**
>
> In stage 1, the choice to refrain from utilizing a pre-trained model is made to ensure a fair comparison with prior research. This decision does not imply that our approach is incapable of leveraging pre-trained models. Notably, MASS does provide pre-trained models for developers, although we opted not to use them. It is important to highlight that employing pre-trained models in stage 1 is a feasible option.
>
> **3. Concerns on the datasets**
>
>  In our work, to ensure a fair comparison, the datasets and the amount of data used at each stage are strictly consistent with the baseline. For Stage 1, a monolingual corpus from WMT with a size of 10.145 million was used. In Stage 2, a monolingual image-text pair dataset from Multi30k and Mscoco, with a size of 75,000, was utilized. In this step, data from both datasets were evenly distributed between the source and target languages to ensure a uniform data distribution, consistent with other UMMT baselines.
>
> It is crucial to emphasize that the datasets and data volumes used at each stage are fixed, and there is no arbitrary allocation. The even distribution of Multi30K and MsCOCO in Stage 2 aligns consistently with other UMMT baselines. The WMT data utilized in Stage 1 is complete, and the text data from Stage 2 is employed in Stage 3, maintaining full consistency with the baseline throughout.
>
> **4. Concerns on Analysis**
>
> For the retrieval task, we did not specifically demand that the MASS model possess retrieval capabilities. Our focus was on conducting a comparison between the performance before and after cross-modal training to illustrate how cross-modal training enables the model to learn cross-modal alignment. Section 5.4 primarily analyzes the impact of different noise levels on the experimental results. It's important to note that we did not emphasize this as our core contribution. In Section 5.5, we selected two representative examples for qualitative illustration. Due to space constraints, we were unable to present more sentences. We will address this by adding an appendix in the final version to provide additional examples.
>
> **5. Concerns on comparisons to other systems are not current**
>
> Firstly, comparing our model to NLLB, M2M100 is unfair as they were trained using a substantial amount of parallel corpora. Secondly, multilingual translation and unsupervised machine translation are two distinct machine translation tasks, making a direct comparison impractical. Lastly, it is crucial to emphasize that, in Stage 1, any encoder-decoder-based pre-trained model could be adopted, including LLMs like FlanT5. However, due to hardware limitations (we only have 20GB 3090s), we couldn't conduct experiments on a larger scale. We will explore further with more extensive computational resources in the future.

---

> ### Author Response · Authors · 2023-11-15
> **Response to Reviewer fBBj (2/2)**
>
> **6. Answer to Q1**
>
> In Section 2, we provide a detailed overview of the roles and purposes of the three stages. The Language Modeling (LM) stage aims to enable the model to learn monolingual representations. The Initialization stage is designed to equip the model with a rudimentary translation capability, offering a good initial value for subsequent iterative back-translation. The Iterative Back-Translation stage guides the model to learn the alignment between the source and target languages.
>
> Certainly, the Initialization stage is not strictly necessary, similar to how iteration can start with a randomly chosen initial value. However, opting for a random initial value often leads to lower-quality iterative results and less training stability. Initialization, on the other hand, provides a solid starting point for iteration, resulting in improved iterative outcomes and enhanced training stability. This underscores the importance of the Initialization stage.
>
> **7. Answer to Q2, Q3, Q4**
>
> Thank you for pointing out errors and inaccuracies in our expression and writing. We apologize for any confusion this may have caused. We will diligently address these issues and rectify the errors in the upcoming version.
>
> **8.  Answer to Q5**
>
> The main reason for this is to reduce the accumulation of errors in Stage 3: iterative back-translation. If a unified decoder is used and translation of wrong language occurs during the back-translation process, it will accumulate in the iteration, thereby impacting the overall model training. This could result in poor training outcomes, and in some cases, the model fails to converge.
>
> **9.  Answer to Q6**
>
> Using [cls] is certainly feasible, and it's a very good suggestion. It's highly likely to yield better results. Thank you very much for bringing this up. We will further explore and experiment with this suggestion to observe its effects.
>
> **10.  Answer to Q7**
>
> The issues you pointed out are typically present in traditional Multimodal Machine Translation (MMT). In traditional supervised MMT tasks, there is ample textual information, and the model's utilization of image information is limited, with images playing a relatively minor role. However, in Unsupervised Multimodal Translation (UMMT), the absence of textual information makes it challenging. If images are replaced with noise or chosen randomly, contrastive learning becomes impossible, and learning cross-modal alignment becomes unfeasible. This highlights the crucial role played by images in this context, rather than better fitting to the distribution of sequence lengths.
>
>
> Thank you again for your valuable time and effort in the review. If our answer solves your problems precisely, we would appreciate it if you could reassess our work. If you have any further questions, please don't hesitate to reach out to me for discussion.

---

> ### Comment · Reviewer_fBBj · 2023-11-16
> **Initial Response**
>
> Thank you for your response --- it is clear I have made an error in assumptions when reviewing the paper. I will reevaluate the paper with the clarifications you have provided and edit my review in a few days time. Please accept my apologies for this error.

---

> ### Author Response · Authors · 2023-11-16
> **Thank you for your prompt response！**
>
> Thank you for your prompt response. We are delighted that we were able to clarify some misunderstandings regarding the paper. I appreciate the time and effort you have invested in this matter. Looking forward to receiving your further response soon. Thanks again!

---

> ### Comment · Reviewer_fBBj · 2023-11-19
> **Additional response.**
>
> I have revised my review in some ways to address weaknesses which are not actually weaknesses. However, I still have several concerns:
>
> [Minor]
> - Final sentence of the second paragraph in 1 is not grammatical and needs revision.
> - Typo for “Vision” in Figure 1.
>
> [Major]
> - My review and other reviewers have identified that your error analysis is anecdotal and your response has addressed that you will include more examples. The number of examples is not the issue here. The issue is that you have produced summary judgements of error analysis based on too few datapoints. Even for a case study, additional discussion is needed to know if this case study represents wider trends.
> There are several directions you could address which are *not* including more examples:
> 	- Are the examples given indicative of wider trends? How many examples in the test set fail due to this exact error?
> 	- How many examples did you choose to analyse for this analysis? Did you randomly sample some percentage of failed outputs? How many did you sample? This needs further contextualisation.
> 	- I do not need to see more examples, I need to see if these examples are indicative of wider errors and behaviours of this system. Can you marry this qualitative discussion with a quantitative discussion of how widespread the qualitative effects are?
>
> - I still consider your justification for Initialisation to be circular: you pointed me back to “The initialization equips the model with a certain level of coarse-grained translation ability, jump-starting the iterative back-translation process. “. This is not what this stage *is*, this is what this stage *does*. Please revise to state what Initialisation *is* — much confusion on my behalf (and other reviewers) seems to stem from this error in communication.
>
> - It is still confusing to me why you can use ViT for image understanding but the UMMT system must be trained from scratch. Can you expand on this discussion somewhere? Can you add the pre-trained component of MASS to verify the effect of pre-training? If MASS has this element, why is it unfair to use it here?
>
> - Similarly, it might be helpful to address why the comparisons I initially listed are unfair/not comparable. They could also be provided somewhere in an appendix as an upper-bound (to contrast what is possible with UMMT vs. Other types of MT).

---

> ### Author Response · Authors · 2023-11-20
> **Further Response to Reviewer fBBj （1/2）**
>
> Thank you for conducting a thorough reassessment of our paper; this is immensely helpful for us. We will provide further responses to your latest questions and hope to address any uncertainties you may have.
>
> **1. Concerns on Case Study**
>
> First and foremost, we sincerely appreciate the suggestion you made from this perspective, which was something we hadn't considered before. After thorough discussion and consideration on our part, we indeed recognize the existence of the issues you pointed out. The original intention behind establishing the Case Study was to provide readers with an intuitive qualitative analysis. However, drawing certain conclusions solely based on the BLEU and METEOR values, the two available quantitative analysis metrics, lacks rigor.
>
> Based on this, we will make modifications to the Case Study section to highlight examples of translations as the primary focus and remove some conclusive descriptions. Additionally, we will supplement the experimental analysis section with experiments supported by quantitative analysis, such as using the Singular Value Gap [1] to measure the alignment levels of different languages before and after applying our method. The specific results are provided below for your reference (lower score means two languages are closer).
>
> **Table 1 Singular Value Gap**
> |                    |       |        |
> |:------------------:|:-----:|:------:|
> |                    | EN-DE | EN-FR  |
> | Before Contrastive | 63.3  | 4.7   |
> | After Contrastive  | 2.7   | 0.1    |
>
> **2. Concerns on the definition of Initialization**
>
> We sincerely apologize for the misunderstanding in our previous responses. Firstly, it should be clarified that the definitions of the elements in Unsupervised Neural Machine Translation (UNMT), namely Language Modeling, Initialization, and Iterative Back-Translation, were not arbitrarily summarized by us. We referenced the definitions and explanations of UNMT in "Phrase-Based & Neural Unsupervised Machine Translation."[2]
>
> In particular, the definition of initialization is as follows: **"Given the ill-posed nature of the task, model initialization expresses a natural prior over the space of solutions we expect to reach, jump-starting the process by leveraging approximate translations of words, short phrases, or even sub-word units."**  Initialization is not a specific task name but rather a step in the UNMT training process, an integral part of UNMT. For example, the training of a Language Model also involves pre-training and fine-tuning steps, and there are various methods available for each. Initialization follows a similar pattern; it is a step in UNMT training that can be accomplished using methods such as bilingual dictionaries or our contrastive learning approach. We hope this explanation provides a clearer understanding of the definition.
>
> **3. Concerns on the using of pre-training model of MASS**
>
> The pre-trained models provided by MASS release are trained on 5 million words of monolingual data from WMT. In contrast, the baselines we are comparing, such as UMMT[3] and PVP[4], are trained on 10 million words of WMT monolingual data. To maintain consistency with these baselines, we opted to use 10 million words of WMT monolingual data during the Language Modeling phase. In terms of data volume, our model uses more data than the pre-trained model provided by MASS, and given the simplicity of our training approach, conducting our own pre-training is entirely feasible. Therefore, this decision is based on the need for consistency with other baselines and the feasibility of conducting our pre-training, rather than relying on the pre-trained model provided by MASS. Additionally, UMMT and PVP both use the pre-trained vision encoder, as a result, we use a pre-trained ViT here.
>
> *[1] The Secret is in the Spectra: Predicting Cross-lingual Task Performance with Spectral Similarity Measures. EMNLP 2020*
>
> *[2] Phrase-Based & Neural Unsupervised Machine Translation. EMNLP 2018*
>
> *[3] Unsupervised Multi-modal Neural Machine Translation. CVPR 2019*
>
> *[4] Unsupervised Multi-modal Neural Machine Translation with Pseudo Visual Pivoting. ACL 2020*

---

> ### Author Response · Authors · 2023-11-20
> **Further Response to Reviewer fBBj （2/2）**
>
> **4. Concerns on whether it is comparable to NLLB, M2M100**
>
> Assessing the fairness of a comparison between two models requires a comprehensive consideration of specific tasks, model sizes, and the datasets used for training. Firstly, the models you mentioned, such as M2M100, primarily focus on multilingual machine translation rather than unsupervised machine translation. Multilingual translation emphasizes translation between different languages, while unsupervised translation concentrates on the unsupervised nature of the method, meaning translation without the involvement of parallel corpora.
>
> Concerning datasets, multilingual models typically leverage abundant parallel corpora for training, which contrasts with the goal of unsupervised machine translation aiming to explore model performance in the absence of parallel data. If a comparison is necessary, it might be essential to fine-tune the unsupervised model using parallel corpora to ensure a fair evaluation in a few-shot scenario. Therefore, ensuring fairness in comparison requires a detailed consideration of task definitions, model configurations, and training data, among other factors.
>
> Certainly, the perspective you mentioned is also very valuable. Exploring the effectiveness of our method on larger-scale multilingual models is something we are genuinely interested in. Our approach might offer a potential solution to mitigate off-target issues in multilingual translation. We will conduct further research on these related issues in the future.
>
> **5. Grammar and spelling errors**
>
> We deeply apologize for the occurrence of such errors in the article and appreciate your pointing them out. We will promptly address and correct them in the next version.
>
> Finally, we sincerely appreciate the time and effort you have dedicated to the review. We hope our further responses can better address your questions. If you have any additional concerns or uncertainties, please do not hesitate to reach out to us. Thank you very much!

---

### Official Review · Reviewer_rLq9 · 2023-10-31

**Soundness:** 3 good
**Presentation:** 3 good
**Contribution:** 3 good
**Rating:** 6
**Confidence:** 4

**Summary:**

This paper presents a novel unsupervised multi-modal machine translation method that leverages monolingual image-text pairs as pivots to learn a shared source-target language space for better initialization through contrastive learning. Experimental results show that this technique leads to a better translation model that outperforms both text-only and multi-modal baselines on machine translation tasks.

**Strengths:**

1. The authors clearly described the background and motivations needed to understand the proposed unsupervised multi-modal machine translation model.
2. The proposed method relies on cross-modal contrastive learning to achieve sentence-level and token-level alignment. This approach results in a strong source-target alignment in the shared space for better initialization before the back-translation step.
3. The authors showed the generalization potential of the proposed model in an out-of-domain experiment.

**Weaknesses:**

1. There are some typos and grammatical errors (I missed some but here is an example. Kindly check and make the necessary corrections):
a. "Therefore, we propose a novel unsupervised multi-modal method to achieve better initialization. method semantically aligns source-target languages into a shared latent space through contrastive learning, using images as pivots"
2. This method heavily relies on the assumption that the two source and language spaces are approximately isomorphic.  What if the two spaces are not isomorphic?

**Questions:**

1. What happens to your model when the two spaces are not isomorphic?
2. A more comprehensive experiment on more languages is needed. The shared space of EN-FR, and EN-DE, is highly approximately isomorphic even under procrustes. Include other languages and report them to make your model more generalizable.
3. No significance testing or error bars on experimental results.
4. By reporting metrics such as singular value gap and effective condition number (see https://aclanthology.org/2020.emnlp-main.186.pdf  and https://openreview.net/forum?id=Nh7CtbyoqV5) could show how the shared representation become after your proposed method is applied. A before and after table should be good to report.
5. Do you apply any normalization technique(s) on the representation to make them isomorphic?

---

> ### Author Response · Authors · 2023-11-19
> **Response to Reviewer  Reviewer rLq9**
>
> We sincerely thank the reviewer for the constructive and helpful feedback. We hope the following responses would help address your concerns.
>
> **1. Concerns on isomorphic spaces**
>
> Firstly, we did not assume any necessity for isomorphism between the source languages, nor did we employ any normalization methods that would enforce convergence towards isomorphism among distant languages. However, the suggestions you provided from this perspective are highly valuable. Therefore, we will conduct further analysis based on your input.
>
> Constrained by the dataset, both previous works and our work have conducted experiments in English, German, and French. For English and French, there is a substantial amount of shared vocabulary, indicating a higher degree of similarity. In the case of English and German, the differences between them are relatively greater, but they still belong to the same language family. Therefore, we chose to conduct experiments with Czech, a language that does not belong to the same language family as English. English belongs to the Indo-European language family, while Czech belongs to the West Slavic language group. The substantial differences between them fulfill the criteria for low similarity. You can refer to Table 1 below for Czech experimental results.
>
> **Table 1 Results on Multi30K EN-CS Flickr2016 set**
> |      |           |        |       |        |           |        |       |        |      |         |
> |:----:|:---------:|:------:|:-----:|:------:|:---------:|:------:|:-----:|:------:|:----:|:-------:|
> |      | Test_2016 |        |       |        | Test_2018 |        |       |        | Avg  |         |
> |      | en-cs     |        | cs-en |        | en-cs     |        | cs-en |        |      |         |
> |      | BLEU      | METEOR | BLEU  | METEOR | BLEU      | METEOR | BLEU  | METEOR | BLEU | METEOR  |
> | MASS | 20.1      | 23.9   | 27.1  | 29.3   | 16.1      | 21.2   | 22.3  | 27.1   | 21.4 | 25.4    |
> | Ours | **24.2**      | **26.4**   | **30.8**  | **32.2**   | **20.0**        | **24.1**   | **26.6**  | **30.4**    |**25.4** | **28.3**    |
>
> **2. Concerns on singular value gap and effective condition number**
>
> Thank you very much for bringing up these metrics. We can use these metrics to quantitatively represent how the shared representation become after your proposed method is applied. Therefore, we have reported the status of **Effective Rank, Effective Condition Number,  Singular Value Gap** in the table 2, 3, 4 below.
>
> **Table 2  Effective Rank**
> |                    |     |     |      |
> |:------------------:|:---:|:---:|:----:|
> |                    | EN  | DE  | FR   |
> | Before Contrastive | 207 | 225 | 207  |
> | After Contrastive  | 224 | 232 | 222  |
>
> **Table 3 Effective Condition Number**
> |                    |      |      |       |
> |:------------------:|:----:|:----:|:-----:|
> |                    | EN   | DE   | FR    |
> | Before Contrastive | 24.3 | 19.4 | 24.3  |
> | After Contrastive  | 16.7 | 17.4 | 17.1  |
>
>
> **Table 4 Singular Value Gap**
> |                    |       |        |
> |:------------------:|:-----:|:------:|
> |                    | EN-DE | EN-FR  |
> | Before Contrastive | 63.3  | 4.7   |
> | After Contrastive  | 2.7   | 0.1    |
>
>
> From the experimental results, it can be observed that before contrastive learning, the singular value gap between German and English is larger compared to English and French. This indicates a greater dissimilarity between English and German (63.3) than between English and French (4.7). Through contrastive learning, the singular value gaps in both English-German and English-French pairs significantly reduced, and the effective condition number also decreased relatively. This further emphasizes the effectiveness of our method.
>
> **3. Concerns on normalization technique(s) on the representation**
>
> It is important to emphasize that we did not apply any normalization technique on the representation to make them isomorphic. The learning of the shared representation space relies entirely on contrastive learning.
>
> **4. Concerns on significance testing or error bars**
>
> We sincerely apologize for the absence of significance testing or error bars on experimental results. Due to the complexity of our experiments involving multiple datasets and language directions, supplementing this specific experiment requires additional time. We will include this in the final version of our paper.
>
> **5. Concerns on  typos and grammatical errors**
>
> Thank you for pointing out errors and inaccuracies in our expression and writing. We apologize for any confusion this may have caused. We will diligently address these issues and rectify the errors in the upcoming version.
>
> Thank you again for your valuable time and effort in the review. If our answer solves your problems precisely, we would appreciate it if you could reassess our work and raise the score. If you have any further questions, please don't hesitate to reach out to me for discussion.

---

> ### Author Response · Authors · 2023-11-22
> **Looking forward to further feedback**
>
> Dear Reviewer rLq9，
>
> I hope this message finds you well. We greatly value your insights and would appreciate your feedback on our submission. We have provided a detailed explanation and additional experimental results for your query. May we inquire if it addressed your concerns, or do you have any further questions? As we approach the rebuttal deadline, if possible, could you kindly provide your comments at your earliest convenience? Your input is crucial to our revision process.
>
> Thank you for your time and consideration.

---

> ### Comment · Reviewer_rLq9 · 2023-11-22
> **Comments after rebuttal**
>
> Thank you for your response and for conducting the additional experiments I requested. For this reason, I am willing to revise my score.  However, it seems the next higher score available is an 8. But I highly recommend this paper. Excellent work!

---

> ### Author Response · Authors · 2023-11-22
> **Thank you for your strong recommendation!**
>
> Thank you for your prompt response and for reviewing the additional experiments. I'm grateful for your willingness to revise the score. We understand the constraints regarding the scoring system,  and we are grateful for your recommendation despite this limitation. However, given our current standing on the borderline, each point holds significant value for us. We sincerely hope for the possibility of achieving a higher score. If there are any specific aspects or information that could contribute to this, please let us know. Thanks again for your thoughtful consideration!

---

### Official Review · Reviewer_QtWc · 2023-11-01

**Soundness:** 3 good
**Presentation:** 3 good
**Contribution:** 3 good
**Rating:** 5
**Confidence:** 5

**Summary:**

This paper presents an innovative approach to Unsupervised Multi-modal Machine Translation (UMMT) that leverages images as language-agnostic signals. The authors introduce cross-modal contrastive learning at both sentence-level and token-level to achieve cross-lingual alignment and enhance translation performance. Experimental results demonstrate that the proposed method surpasses state-of-the-art UMT and UMMT systems in terms of BLEU and METEOR scores.

**Strengths:**

1.	The introduction of the visual modality as a language-agnostic signal is a novel approach that holds the potential to enhance the effectiveness of UMT systems.
2.	The extensive experimental evaluation conducted on multiple datasets demonstrates the superiority of the proposed method compared to other UMT and UMMT systems.

**Weaknesses:**

1.	This work primarily evaluates English-German and English-French translations, which are typically high-resource translation tasks. It would be valuable to see an evaluation of real low-resource languages to better gauge the method's effectiveness in such scenarios.
2.	The paper should provide insights into the effectiveness of the alignment method when applied to languages with low similarity. This would offer a more comprehensive understanding of its performance across various language pairs.
3.	In comparison to methods that rely on bilingual dictionaries to enhance alignment, such as denoising synthetic code-switched data, the paper should discuss whether the introduction of images offers clear advantages or if these two approaches are complementary.
4.	The authors should consider discussing whether there have been more recent developments in text-only unsupervised translation methods, as this would help place their approach in the context of the latest advancements in the field.

**Questions:**

While the authors demonstrate superiority over existing systems on high-resource language pairs (en<->de and en<->fr), they should explore the effectiveness of real low-resource languages

---

> ### Author Response · Authors · 2023-11-19
> **Response to Reviewer QtWc (1/2)**
>
> We sincerely thank the reviewer for the constructive and helpful feedback. We hope the following responses would help address your concerns.
>
> **1. Concerns on real low-resource languages**
>
> Constrained by the dataset, both previous works and our work have conducted experiments in English, German, and French. However, it is necessary and valuable to validate our approach on truly low-resource languages. Therefore, we conducted additional experiments using a Czech language dataset from the Multi30K Test2016 and Test2018 datasets.
> Our method demonstrates a substantial improvement compared to the text-only baseline, with an average BLEU score increase of 4.0 and an average METEOR score increase of 2.9 across all the language directions. This indicates that our method also performs well on real low-resource languages.
>
> **Table 1 Results on Multi30K EN-CS Flickr2016 set**
> |      |           |        |       |        |           |        |       |        |      |         |
> |:----:|:---------:|:------:|:-----:|:------:|:---------:|:------:|:-----:|:------:|:----:|:-------:|
> |      | Test_2016 |        |       |        | Test_2018 |        |       |        | Avg  |         |
> |      | en-cs     |        | cs-en |        | en-cs     |        | cs-en |        |      |         |
> |      | BLEU      | METEOR | BLEU  | METEOR | BLEU      | METEOR | BLEU  | METEOR | BLEU | METEOR  |
> | MASS | 20.1      | 23.9   | 27.1  | 29.3   | 16.1      | 21.2   | 22.3  | 27.1   | 21.4 | 25.4    |
> | Ours | **24.2**      | **26.4**   | **30.8**  | **32.2**   | **20.0**        | **24.1**   | **26.6**  | **30.4**    |**25.4** | **28.3**    |
>
> To further supplement our findings, we conducted experiments on English-to-German and English-to-French translation under low-resource settings, specifically using only 75,000 monolingual data samples.
>
> **Table 2 Results on low-resource setting (75k data only).**
> |      |       |        |       |        |       |        |       |        |      |         |
> |:----:|:-----:|:------:|:-----:|:------:|:-----:|:------:|:-----:|:------:|:----:|:-------:|
> |      | en-de |        | de-en |        | en-fr |        | fr-en |        | Avg  |         |
> |      | BLEU  | METEOR | BLEU  | METEOR | BLEU  | METEOR | BLEU  | METEOR | BLEU | METEOR  |
> | MASS | 18.3  | 35.8   | 19.7  | 23.4   | 24    | 41.5   | 23.9  | 27     | 21.5 | 31.9    |
> | Ours | **27.5**  | **47.6**   | **32.3**  | **32.2**   | **36.7**  | **54.6**   | **36.9**  | **34.3**   | **33.6** | **42.2**    |
>
> The experimental results indicate that in simulated low-resource scenarios, our method continues to yield significant improvements. In contrast, text-only method without large-scale monolingual pretraining experiences a severe performance drop, and the model convergence during training is notably slower.
>
> **2. Concerns on languages with low similarity**
>
> For English and French, there is a substantial amount of shared vocabulary, indicating a higher degree of similarity. In the case of English and German, the differences between them are relatively greater, but they still belong to the same language family. Therefore, we chose to conduct experiments with Czech, a language that does not belong to the same language family as English. English belongs to the Indo-European language family, while Czech belongs to the West Slavic language group. The substantial differences between them fulfill the criteria for low similarity.
> You can refer to Table 1 above for Czech experimental results.

---

> ### Author Response · Authors · 2023-11-19
> **Response to Reviewer QtWc (2/2)**
>
> **3. Concerns on comparison to methods that rely on bilingual dictionaries**
>
> Actually, we have already compared our approach with the method relying on bilingual dictionaries in our paper. In Section 5.2 (Translation Quality), we compared the pre-back-translation results. Here, "MUSE"[1] represents the method initialized with a bilingual dictionary. Regarding the post-back-translation results, in the Baselines, the UMT[2] method employs the initialization method based on a bilingual dictionary. The results obtained by our method are significantly superior to UMT method.
>
> Additionally, obtaining a high-quality bilingual dictionary requires a significant amount of additional data expenses, such as through manual annotation or extraction from bilingual parallel data. This is not aligned with the objectives of Unsupervised Neural Machine Translation (UNMT). Relatively speaking, our method achieves high-quality translation with lower data collection costs as it only requires monolingual text-image pairs.
>
> **4. Concerns on more recent text-only unsupervised translation methods**
>
> Recently, many state-of-the-art text-only translation models are based on larger-scale models for multilingual translation. These models are not entirely unsupervised; their training often involves extensive parallel corpora. In these models, zero-shot tasks are common for languages without parallel data. Our method can be operated at the representation level, narrowing the gap between representations of different languages. This helps alleviate off-target issues (outputting in the wrong language) in multilingual translation models, thereby improving overall performance. Consequently, our approach is equally applicable to large language models. In future explorations, we plan to delve into related research. Thank you for your valuable suggestions!
>
> Thank you again for your valuable time and effort in the review. If our answer solves your problems precisely, we would appreciate it if you could reassess our work and raise the score. If you have any further questions, please don't hesitate to reach out to me for discussion.
>
> *[1] Word Translation without Parallel Data. ICLR 2018*
>
> *[2] Unsupervised Machine Translation using monolingual corpora only. ICLR 2018*

---

> ### Author Response · Authors · 2023-11-22
> **Looking forward to further feedback**
>
> Dear Reviewer QtWc，
>
> I hope this message finds you well. We greatly value your insights and would appreciate your feedback on our submission. We have provided a detailed explanation and additional experimental results for your query. May we inquire if it addressed your concerns, or do you have any further questions? As we approach the rebuttal deadline, if possible, could you kindly provide your comments at your earliest convenience? Your input is crucial to our revision process.
>
> Thank you for your time and consideration.

---

### Official Review · Reviewer_QpzC · 2023-11-03

**Soundness:** 2 fair
**Presentation:** 3 good
**Contribution:** 2 fair
**Rating:** 5
**Confidence:** 3

**Summary:**

Motivated by the thoughts that different languages share a common visual description, this paper proposes a novel unsupervised multi-modal machine translation method using images as pivots. Specifically, sentence-level and token-level alignment are achieved by contrastive learning. Experiments show that its method achieves improvements over other methods.

**Strengths:**

The writing is clear and brief, making readers easy to understand.

**Weaknesses:**

1. The text-image pairs need annotation, which means that the unsupervised machine translation needs cross-modality annotation, which is more expensive than text-only annotation.
2. In Section 3.3, word is the text token. What does an image token represent? why text token and image token in different position (i ≠ j) is regarded as negative examples, and why in same position can be regarded as positive examples.

**Questions:**

1. In Section 3.3, the word is the text token. What does an image token represent?
2. In section 3.3, why text tokens and image tokens in different positions (i ≠ j) is regarded as negative examples, and why in the same position can be regarded as positive examples? In text-only translation tasks, the same position of the src sentence and tgt sentence always do not refer to the same thing. In other words, the mapping of image tokens and language tokens is simply one-to-one position mapping.
3. Do you only visualize 6 examples in Figure 2? Where are the 6 examples from? Is it sufficient to support that your approach is truly successful?
4. Why IWSLT data is regarded as out-of-domain data compared with the data you use for initialization which includes MsCOCO and Multi30K? Can you prove the domain mismatch in the two kind of datasets?

---

> ### Author Response · Authors · 2023-11-15
> **Response to Reviewer QpzC (1/2)**
>
> We sincerely thank the reviewer for the constructive and helpful feedback. We hope the following responses would help address your concerns.
>
> **1. Concerns on the cost of cross-modality annotation**
>
> In Unsupervised Multi-modal Machine Translation (UMMT), the correspondence between source language and target language is missing, and visual information serves as additional data to compensate for this absence. Although there is costly for data collection, our cross-modal model has achieved a significant improvement in performance compared to the text-only baseline. Additionally, we only utilize monolingual image-text annotations,  it is easier to recruit monolingual annotators to describe an image than to find multilingual translators to translate sentences.which results in relatively lower data collection costs.
>
> **2. Concerns on the image token and token-level contrastive learning**
>
> We utilize the **Vision Transformer (ViT)** as our image encoder. In Vision Transformer,  the input image is divided into fixed-size non-overlapping patches. Each patch is treated as a token. After that, each patch is linearly embedded to flatten it into a 1D vector. These vectors serve as the initial input tokens for the transformer model. Finally, it encodes the image to a sequence $v = (v_0, v_1, ..., v_m)$, where $v_0$ is the special $[class]$ token and the others are the representation of image patches. For example, a single image is divided into 9 patches through a 3x3 grid. Each patch serves as a token, and when combined with a special [class] token, it forms a sequence with a length of 10 as input to the model. Consequently, a vector of dimensions (1, 10, 512) is obtained after processing through the model.
>
> Therefore, each image token represents the vector representation of a patch of the image. We utilize a selective attention module to filter out image region representations related to the semantics of each text token from multiple patch representations, and unify the sequence length. Consequently, the attention sequences filtered for each token correspond one-to-one. Instantly, $w_1$ and $v^t_1$ is positive examples, $w_1$ and $v^t_1$ is negative examples. The token-level contrastive learning is trained between sequence w and $v^t$ but not $w$ and $v$.
>
> A core module here is the **Selective Attention** module, which filters image information based on each text token. As a result, beyond positional correspondence between sequences, there exists a strong semantic relevance. Due to limitations in the length of the paper, we don't present the detailed image feature extraction process in the original text. You can refer to Figure 1 of <On Vision Features in Multimodal Machine Translation>[1] and <An image is worth 16x16 words: Transformers for image recognition at scale>[2] for a more in-depth understanding. It will greatly aid your comprehension.
>
> **3. Concerns on the visualization**
>
> We randomly pick 6 examples from the Test2016 set of Multi30k on EN-DE direction. and due to space constraints, additional examples couldn't be included in a single diagram. For a more comprehensive analysis, please refer to the text-to-image retrieval section (Sec 5.2), where you'll find that the overall matching performance between images and text has considerably improved across the entire test set. Therefore, Figure 2 only serves as a visual example specifically for the text-to-image retrieval part, aiming to enhance reader understanding.
>
>
> *[1] On Vision Features in Multimodal Machine Translation. ACL 2022*
>
> *[2] An image is worth 16x16 words: Transformers for image recognition at scale. ICLR 2021*

---

> ### Author Response · Authors · 2023-11-15
> **Response to Reviewer QpzC (2/2)**
>
> **4. Concerns on IWSLT dataset**
>
> The Multi30k and MsCOCO dataset is primarily designed for image captioning tasks. It consists of images paired with descriptions, making it suitable for tasks related to visual understanding and language generation. The IWSLT dataset, on the other hand, is created for speech translation tasks. It includes spoken language data in various languages and is commonly used for evaluating the performance of machine translation systems for spoken content. We provide some examples below to help you understand.
>
> **Multi30k:**
>
> [English]: A man in a blue shirt is standing on a ladder cleaning a window.
>
> [German]: Ein Mann in einem blauen Hemd steht auf einer Leiter und putzt ein Fenster.
>
> [English]: A group of people are sitting on a wall and playing musical instruments.
>
> [German]: Eine Gruppe von Menschen sitzt auf einer Mauer und spielt Musikinstrumente.
>
> **IWSLT:**
>
> [English]: Can you help me find my keys? I think I left them on the kitchen counter.
>
> [German]: Kannst du mir helfen, meine Schlüssel zu finden? Ich glaube, ich habe sie auf der Küchentheke liegen lassen.
>
> [English]: The conference will be held in the main auditorium, starting at 9:00 AM.
>
> [German]: Die Konferenz findet im Hauptauditorium statt und beginnt um 9:00 Uhr.
>
> These examples illustrate typical sentences from each dataset, showcasing the diversity of language and content in Multi30k (image captioning) and IWSLT (text translation).
>
>
> Thank you again for your valuable time and effort in the review. If our answer solves your problems precisely, we would appreciate it if you could reassess our work. If you have any further questions, please don't hesitate to reach out to me for discussion.

---

> ### Author Response · Authors · 2023-11-22
> **Looking forward to further feedback**
>
> Dear Reviewer QpzC，
>
> I hope this message finds you well. We greatly value your insights and would appreciate your feedback on our submission. We have provided a detailed explanation for your query. May we inquire if it addressed your concerns, or do you have any further questions? As we approach the rebuttal deadline, if possible, could you kindly provide your comments at your earliest convenience? Your input is crucial to our revision process.
>
> Thank you for your time and consideration.

---

### Author Response · Authors · 2023-11-20
**Looking forward to further feedback**

Dear Reviewers,

Thanks again for your valuable comments. We have responded to all reviewers' comments. We hope our responses are useful to address your concerns. As we are approaching the end of the rebuttal phase, we would be grateful to receive further feedback from you, and we would be happy to address any remaining concerns.

---

### Author Response · Authors · 2023-11-21
**Revised paper uploaded**

- We sincerely thank all reviewers for their time and efforts. Now we have uploaded our revised paper with the following revision:

    - **We added results on En->CS and CS->EN in Sec 5.4 to show the effectiveness of the alignment method when applied to languages with low similarity and the performance on real low-resource language.**

    - We reported the singular value gap and effective condition number in Appendix B to show how the shared representation become after your proposed method is applied.

    - We added the experiment on low-resource settings in Appendix C.

    - We moved the noise analysis to Appendix A.

    - We have corrected all spelling and grammar errors identified by the reviewers.

    - we maked modifications to the Case Study section to highlight examples of translations as the primary focus and remove some conclusive descriptions.

**As we are approaching the end of the rebuttal phase, we would be grateful to receive further feedback from you, and we would be happy to address any remaining concerns.**

---

### Meta-Review · Area_Chair_tK6L · 2023-12-15

**Metareview:**

This paper presents an innovative approach to Unsupervised Multi-modal Machine Translation (UMMT) that leverages images as language-agnostic signals. The authors introduce cross-modal contrastive learning at both sentence-level and token-level to achieve cross-lingual alignment and enhance translation performance. Experimental results demonstrate that the proposed method surpasses state-of-the-art UMT and UMMT systems in terms of BLEU and METEOR scores.
The introduction of the visual modality as a language-agnostic signal is a novel approach and the extensive experimental evaluation conducted on multiple datasets demonstrates the superiority of the proposed method compared to other UMMT approaches. During the rebuttal period, the author added more experiments for cs<->en to address concerns and questions of the reviewers to demonstrate that their approach is also working for less similar languages.
The approach is interesting, novel and well presented. However, there are major concerns regarding the usefulness of their experimental results. The experimental setup is a bit outdated, especially for the language pairs they run their experiments on. They train a very small transformer model from scratch instead of initializing from a pre-trained model. This might be appropriate for real low-resource language pairs, but not for high or mid-resource languages where pre-trained models are available. This hurts the impact of the paper in its current form as it is less clear how this work could be interpreted/extended by a current reader. I encourage the reader to read [1] about this topic and add experiments when initializing the model with a pre-trained model. Related to that, for language pairs where in reality no parallel data is available, it is unclear how much image labeled data is available or can be generated. This would be worth to discuss in the paper.

[1] https://aclanthology.org/2020.wmt-1.68/

**Justification For Why Not Higher Score:**

The experimental results are not insightful as they were conducted on language pairs where a massive number of parallel sentences and pre-trained models exist.

**Justification For Why Not Lower Score:**

N/A

---

### Decision · Program_Chairs · 2024-01-16

Reject